# Differentially Expressed Genes and Signalling Pathways Regulated by High Selenium Involved in Antioxidant and Immune Functions of Goats Based on Transcriptome Sequencing

**DOI:** 10.3390/ijms24021124

**Published:** 2023-01-06

**Authors:** Xu Wang, Chao Ban, Jia-Xuan Li, Qing-Yuan Luo, Ji-Xiao Qin, Yi-Qing Xu, Qi Lu, Xing-Zhou Tian

**Affiliations:** 1Key Laboratory of Animal Genetics, Breeding and Reproduction in the Plateau Mountainous Region, Ministry of Education, College of Animal Science, Guizhou University, Guiyang 550025, China; 2Institute of Animal Nutrition and Feed Science, Guizhou University, Guiyang 550025, China

**Keywords:** selenium-yeast, oxidative stress, antioxidant and immune, transcriptome sequencing, goats

## Abstract

The objective of this study is to observe the effect of high selenium on the antioxidant and immune functions of growing goats based on transcriptome sequencing. Eighteen goats were randomly divided into three groups: (1) the control (CON) group was fed a basal diet, and (2) the treatment 1 group (LS) and treatment 2 group (HS) were fed a basal diet with 2.4 and 4.8 mg/kg selenium-yeast (SY), respectively. The results indicate that HS treatment significantly (*p* < 0.05) increased the apparent digestibility of either extract and significantly increased (*p* < 0.05) total antioxidant capacity, whereas it significantly (*p* < 0.05) decreased plasma aspartate aminotransferase and malondialdehyde relative to the control group. The LS treatment had significantly (*p* < 0.05) increased glutathione S-transferase and catalase compared to CON. A total of 532 differentially expressed genes (DEGs) between the CON and HS were obtained using transcriptome sequencing. Kyoto Encyclopedia of Genes and Genomes analysis identified upregulated (*p* < 0.05) DEGs mainly related to vascular smooth muscle contraction, alpha-linolenic acid metabolism, biosynthesis of unsaturated fatty acids, the VEGF signalling pathway, and proteoglycans in cancer; downregulated (*p* < 0.05) DEGs mainly related to the NOD-like receptor signalling pathway, influenza A, cytokine-cytokine receptor interaction, haematopoietic cell lineage, and African trypanosomiasis. Ontology analyses of the top genes show that the identified DEGs are mainly involved in the regulation of granulocyte macrophage colony-stimulating factor production for biological processes, the external side of the plasma membrane for cellular components, and carbohydrate derivative binding for molecular functions. Seven genes are considered potential candidate genes for regulating antioxidant activity, including selenoprotein W, 1, glutathione peroxidase 1, glutathione S-transferase A1, tumour necrosis factor, tumour necrosis factor superfamily member 10, tumour necrosis factor superfamily member 8, and tumour necrosis factor superfamily member 13b. The experimental observations indicate that dietary supplementation with 4.8 mg/kg SY can enhance antioxidant and immune functions by improving muscle immunity, reducing the concentrations of inflammatory molecules, and modulating antioxidant and inflammatory signalling pathways in growing goats.

## 1. Introduction

Oxidative stress results from an imbalance between the generation of oxygen-derived radicals and the antioxidant potential of ruminants [1]. Antioxidants, as free radical scavengers, can effectively complement the deficiencies of in vivo antioxidants and play a significant role in maintaining the balance of redox levels. Selenium (Se) is a trace element known to have antioxidant properties and to be involved in healthy immune system activity [2]. Se has roles in immune function and thyroid hormone metabolism and is required for the antioxidant activity of the enzyme glutathione peroxidase (GPX) in animals [3]. Selenium-yeast (SY), produced by growing select strains of *Saccharomyces cerevisiae* in Se-rich media, is a common form of organic food-form Se used in supplementing the dietary intake of this important trace mineral [4]. SY is the most appropriate form of Se to be used as a food supplement because of its lower toxicity and high bioavailability compared with inorganic selenious compounds [5]. Čobanová et al. [6] found that dietary supplementation with SY increases Se levels, enhances GPX and thioredoxin reductase activities, and decreases malondialdehyde content in sheep tissues. Jia et al. [7] demonstrated that feeding SY at less than 2.0 mg/kg reduces thioredoxin reductase activity and decreases thiobarbituric acid reactive substances in the muscle tissue of Tan sheep.

Various studies have shown that Se can improve antioxidant activities and alleviate oxidative stress in the body and that small ruminants have strong tolerance to Se [8,9]. Low Se intake can cause ruminant animals to be prone to numerous problems, including white muscle disease in calves and lambs, pneumonia, scouring, reduced wool production, and reproductive failure [10]. Moreover, microbial digestion in the reticulum and rumen precedes digestion in the abomasum and small intestine, resulting in the much lower absorption of Se in ruminants than in nonruminants [11]. Specifically, the content of dietary Se might be deficient because the Se concentration within forage can be extremely variable, and supplementation of Se in the diet of ruminants may be required [12]. Thus, Se supplementation may enhance antioxidant and immune functions to improve the health conditions of ruminants.

With the rapid development of next-generation sequencing, RNA sequencing (RNA-Seq) provides researchers with a powerful toolbox for characterisation and quantification of the transcriptome and is widely used in studying the regulation of gene expression [13]. For instance, Zhao et al. [14] identified differentially expressed genes (DEGs) between goats treated with and without the antioxidant metformin using RNA-Seq technology and found that metformin regulates testis function, semen quality, antioxidant levels, and autophagy capacity in goats. Moreover, Cheng et al. [15] found that the antioxidant capacity of young ruminants acts as an indicator of health; they performed Kyoto Encyclopedia of Genes and Genomes (KEGG) functional enrichment analysis of their RNA-Seq results and found that the upregulated DEGs are mainly related to the ECM-receptor interaction and axon guidance and that the downregulated DEGs are mainly related to arachidonic acid metabolism, complement and coagulation cascades, and alpha-linolenic acid metabolism in neonatal goats with diarrhoea compared with control goats.

Previous studies have shown that ruminants have a high Se tolerance because Se is less available to ruminants than to nonruminants, and the maximum requirement for Se in ruminant diets is 5 mg/kg [16,17]. For example, Suganthi [18] found that dietary supplementation of 4.5 ppm SY can significantly upregulate selenoprotein P and W1 gene expression in lambs. In addition, in our previous studies, we found that the feeding of high SY has the ability to enhance muscle antioxidant activity and improve rumen fermentation parameters as well as microbes in goats [19,20]. However, the molecular mechanism through which high levels of SY impact the muscle antioxidant activity of growing goats remains unknown. We hypothesised that a high level of Se could modulate related signalling pathways and the expression of related gene to improve antioxidant activity in goats. Accordingly, in the current study, we investigate the mechanism through which high SY impacts the nutrient apparent digestibilities, plasma biochemical and antioxidant activity parameters, and the muscle antioxidant function of growing Qianbei-pockmarked goats based on transcriptome sequencing.

## 2. Results

### 2.1. Apparent Digestibility

Compared to the control group, the inclusion of 4.8 mg/kg SY did not change (*p* > 0.05) the apparent digestibilities of DM, OM, GE, CP, NDF, ADF, Ca, or P, whereas it did significantly (*p* < 0.05) increase EE apparent digestibility (Table 1).

### 2.2. Plasma Biochemical and Antioxidant Activity Parameters

There were no significant differences (*p* > 0.05) for Glu, UN, ALT, TC, TG, Cr, Alb, SOD, GPX, or GSH between the three groups (Table 2). The feeding of SY significantly decreased (*p* < 0.05) AST concentration, whereas it significantly increased (*p* < 0.05) TP content compared with the group without SY. The level of TAC was significantly (*p* < 0.05) higher in the treatments than the control. No significant differences (*p* > 0.05) were observed for GST and CAT between LS and HS treatments, whereas the LS treatment significantly (*p* < 0.05) increased GST and CAT compared with CON. In comparison, kids receiving 4.8 mg/kg SY had significantly (*p* < 0.05) decreased MDA levels relative to the control group.

### 2.3. Muscle Immune Parameters and Oxidation Products

As shown in Table 3, there were no significant (*p* > 0.05) differences in the muscle IgA, IgG, SEPP, IL−1, 8-OHdG, NO, or ·OH levels between the two groups. Compared with the CON group, the inclusion of 4.8 mg/kg SY significantly (*p* < 0.05) increased IgM content, whereas it significantly (*p* < 0.05) reduced the concentrations of NF-κB and O^2−^ in the plasma of growing goats.

### 2.4. Transcriptome Sequencing Results

Transcriptome sequencing data were analysed for 12 muscle samples, and 81.31 gigabytes of raw base data were obtained in this study (Appendix A). A total of 578,203,182 raw reads were obtained, with an average of 3,106,972 ± 896,906 (mean ± standard error of the mean (SEM)) raw reads per sample. The error rate was <0.025%, Q20 was >98%, and Q30 was >94.50%. In addition, the GC content ranged from 51.38% to 53.45%, and the uniquely mapped GC content ranged from 85.57% to 89.06%. Principal component analysis (PCA) of the transcriptome profiles was performed; principal component (PC) 1 was found to account for 29.99% of the total variance, and PC2 for 18.97% of the total variance. Moreover, the CON and HS groups could be clearly discriminated in the PCA plot (Figure 1). In addition, we obtained 532 DEGs between the two groups for transcriptome sequencing (Figure 2; Appendix A). Moreover, a total of 138 upregulated genes (*p* adjust < 0.05) and 394 downregulated genes (*p* adjust < 0.05) were identified in the HS group relative to the CON group.

### 2.5. Kyoto Encyclopedia of Genes and Genomes Enrichment Analysis

The KEGG pathway enrichment (*p* adjust < 0.05) results show that the DEGs were enriched in 247 KEGG pathways, with the top 20 corresponding to enrichment in various pathways, including 2 in cellular processes, 4 in environmental information processing, 8 in human diseases, and 6 in organismal systems (Figure 3; Appendix A).

The upregulated DEGs were enriched in 119 KEGG pathways (Appendix A), and 5 KEGG pathways were significantly enriched (*p* < 0.05), including vascular smooth muscle contraction, alpha-linolenic acid metabolism, biosynthesis of unsaturated fatty acids, VEGF signalling pathway, and proteoglycans in cancer (Figure 4). The downregulated DEGs were enriched in 231 KEGG pathways (Appendix A), and 42 KEGG pathways mainly related to the NOD-like receptor signalling pathway, influenza A, cytokine-cytokine receptor interaction, haematopoietic cell lineage, and African trypanosomiasis were significantly enriched (*p* < 0.05; Figure 5).

### 2.6. Gene Ontology Enrichment Analysis

The gene ontology analysis shows that the identified DEGs are mainly involved in the regulation of granulocyte macrophage colony-stimulating factor production (6 DEGs) for biological processes, the external side of the plasma membrane (21 DEGs) for cellular components, and carbohydrate derivative binding (90 DEGs) for molecular functions (Appendix A).

### 2.7. Expression of Genes Related to Antioxidant Activity

Based on the results of the differential gene expression, GO, KEGG pathway, and gene function analyses, seven genes were considered potential candidate genes for antioxidant activity regulation, which consisted of three upregulated genes, *SEPW1*, *GPX1*, and *GSTA1*, and four downregulated genes, *TNF*, *TNFSF10*, *TNFSF8*, and *TNFSF13B* (Appendix A). Next, we validated the differential expression of these genes via real-time PCR, and the real-time PCR results are consistent with the transcriptome data (Figure 6).

## 3. Discussion

SY has high bioavailability in animals and is an excellent selenium nutritional additive for ruminants [21]. Various studies have found that the inclusion of SY in ruminant diet can improve rumen fermentation parameters and promote the proliferation of lactic acid bacteria and cellulose bacteria, and thus improve the digestion and absorption of feed nutrients [22,23]. With our previous study, we demonstrated that SY could have an effect on the ruminal fluid fermentation parameters, microbial diversity, and microbial metagenome of goats [20]. In the present study, we found that goats receiving 4.8 mg/kg SY had significantly improved EE apparent digestibility, perhaps because SY can provide a certain nutrient for ruminal-fluid-beneficial microorganisms, and thus enhance the ability of rumen microorganisms to digest dietary nutrients. Consistent with our results, Wang et al. [24] found that feeding 0.4 mg/kg SY could increase EE digestibility in Tibetan sheep. Similarly, Morsy et al. [25] demonstrated that SY can improve ruminal fluid fermentation and nutrient apparent digestibility in goats. The cellular oxidative stress response and redox regulation mechanisms must play critical roles in the protection of liver from oxidative damage [26]. Specifically, SY has protective effects on ochratoxin A-induced hepatotoxicity, which could reduce the activity of AST in blood and involve some oxidative-stress-related genes (*Nrf2*, *GLRX2* and *Keap1*) in animals [27]. Thus, we found that SY can decrease plasma AST concentration, suggesting that SY may protect the liver from oxidative stress.

The radical reactions in the body must remain in dynamic balance. Disrupting the equilibrium of the antioxidant system results in oxidative stress if the ability of the antioxidant system to eliminate oxygen radicals is compromised or overexpression of free radicals occurs, resulting in the accumulation of oxidation products in ruminants [28]. Promoting the clearance of free radicals and the antioxidation activity of the antioxidant system may be the main mechanisms through which it regulates and improves the metabolic balance of free radicals to alleviate oxidative stress status and reduce oxidation products [29]. Bioantioxidants can help people maintain a balance of free radicals in the body. Antioxidants are chemicals that can maintain a balance of free radicals and have great value in strengthening the antioxidant status of biological systems [30]. Se, as a powerful antioxidant that can protect other easily oxidised substances, can modulate lipid metabolism and remove excessive free radicals by synthesising Se-related antioxidant enzymes and improve antioxidant activity in animals [31]. Hence, feeding SY could increase plasma TAC content as observed in the present research. One study found that dietary supplementation with SY has the potential to alleviate lipid peroxidation and improve the unsaturated fatty acid profile in the muscle of goats [19]. In ruminants, Se can increase antioxidant status and immune responses and defend against the accumulation of hydroperoxides produced by cellular metabolism [32]. Accordingly, SY feeding decreases plasma MDA and muscle NF-κB and O^2−^ in goats, perhaps because Se can restore normal health by minimising the harmful effects of excessive reactive oxygen species (ROS) production [33]. Consistent with our results, Han et al. [34] showed that the inclusion of Se improves antioxidant status by decreasing blood lipid peroxidation and free radicals in beef cattle fluorosis. Similarly, Sathya et al. [35] demonstrated that supplementation with Se may be beneficial in reducing oxidative stress by decreasing plasma MDA content in dystocia-affected buffaloes.

In a prior study, it was demonstrated that Se affects all aspects of the immune system, such as the development and expression of nonspecific, humoral, and cell-mediated responses [36]. An essential method for improving ruminal fluid microorganisms is supplementation of Se into ruminant rations to increase immune function [37]. Immunoglobulins, which include IgA, IgG, and IgM, are important immune substances in the ruminant body that play important roles in modulating physiological function [38]. Notably, dietary supplementation with Se can result in transfer of Se to muscle and increase antioxidant activity in ruminants. Shi et al. [39] found that dietary supplementation with Se increases Se levels in blood and tissue in addition to growth performance and serum antioxidant activity in growing male Taihang black goats. Accordingly, we found that dietary supplementation with SY improves plasma IgM levels, perhaps because Se-supplemented goats have a better gut microflora composition and immune response [20]. Our results are in agreement with those of Arain et al. [40], who demonstrated that inclusion of Se could significantly increase IgG concentrations in female crossbred goats.

PCA of the transcriptome data in muscle found obvious discrimination between the CON and HS groups. Furthermore, the transcriptional changes were examined at the overall level after SY feeding in goats via KEGG and GO classification analysis. In this study, the KEGG analysis found that the DEGs are mainly involved in vascular smooth muscle contraction and the NOD-like receptor signalling pathway, and the GO analysis found that the DEGs are mainly involved in the regulation of the granulocyte macrophage colony-stimulating factor production pathway, which is related to immunity and antioxidant pathways. Not only does inflammation induce oxidative stress, but oxidative stress also accelerates inflammation through the activation of proinflammatory pathways; for example, the NOD-like receptor signalling pathway is involved in inflammasome activation, and during that process, a series of signalling pathways can be activated, including the NF-κB and AMP-activated protein kinase (MAPK) pathways [41,42]. Indeed, both the NF-κB and MAPK pathways are responsive to ROS activation, resulting in the increased expression of cytokines, chemokines, and other factors involved in the inflammatory response in ruminants [43]. As a source of antioxidants, dietary supplementation with Se has been shown to enrich innate immune responses and cytokine-production-related terms and pathways, such as the NOD-like receptor signalling pathway, NF-κB signalling pathway, and cytokine-cytokine receptor interaction, via KEGG and GO analysis in mice [44]. Hence, we found that Se supplementation inhibited the NOD-like receptor signalling pathway, suggesting that exogenous SY may attenuate oxidative stress injury and decrease inflammatory reactions, which may help relieve inflammation and adjust cytokine levels to increase antioxidant activity.

Selenoproteins function not only as antioxidant enzymes but also in redox signalling and regulation of immune responses because they can reduce toxic ROS to less reactive molecules and regulate intercellular signalling pathways that lead to inflammatory gene expression [43]. The biological functions of Se in organisms are mediated through various selenoproteins in biological systems. Moreover, various selenoproteins play very important roles in key biological functions (antioxidant activity, immunity, etc.) and modulate GPX enzymatic function [45]. Se can react with sulphur and be incorporated into Se-containing amino acids (such as cysteine and methionine) to form the biologically important compounds selenocysteine and selenomethionine [46]. Of note, SY can protect against excessive incorporation of Se into selenoproteins and prevent toxicity mediated through ROS resulting from excessive Se intake [47]. Thus, we found that SY has the ability to improve the abundance of *SEPW1* in the LD muscle of growing goats. Our results are in agreement with those of Suganthi et al. [18], who demonstrated that the inclusion of 1.5 and 4.5 ppm Se increases the abundances of *SEPP* and *SEPW1* mRNA in male lambs.

Se is an essential microelement and a major component of GPX, and it can provide a protective effect against chemical hazards [48]. As a source of antioxidants, Se has strong antioxidant potential, which indicates a possible catalytic protective effect on lipids of Se as a component of GPX due to an electron transfer role of Se in cytochrome systems. Se also protects protein sulphhydryls, and there are possible relationships of Se to cytochromes and oxygen metabolism in ruminants [49]. Oxidative stress biomarkers, such as the proinflammatory cytokine TNF and the free radicals of ROS and reactive nitrogen species, increase and negatively affect body health when ruminants are under conditions of oxidative stress [50]. Selenoproteins play a role in regulating ROS and redox status in nearly all tissues, affecting inflammation and immune responses [10]. Moreover, GPX enzymes destroy hydrogen peroxide and lipid hydroperoxides that may function to prevent oxidative stress, and Se functions in the antioxidant system as an essential component of a family of GPX enzymes [51]. Thus, we found that supplementation with SY can improve *GPX1* and *GSTA1* gene expression in the muscle of goats, perhaps due to GPX being a source of selenoprotein-coding genes and possessing peroxidase activity that provides protection from reactive oxygen and nitrogen species [18]. Consistent with our observations, Çiçek et al. [52] found that the addition of Se could upregulate *SEPW1* and *GPX1* gene expression in the livers of ruminants.

*TNF* is a key regulator of the inflammatory response that can block the action of *TNF* to treat a range of inflammatory conditions, including rheumatoid arthritis, ankylosing spondylitis, inflammatory bowel disease, and psoriasis [53]. Moreover, *TNFSF10*, *TNFSF8*, and *TNFSF13B* are members of the *TNF*-related apoptosis-inducing ligand family and can induce apoptosis through the participation of its death receptor. Se is very important for chemotactic and phagocyte activity and respiratory burst activities, which can decrease inflammatory activity, and the biosynthesis of selenoproteins is disturbed when Se content is decreased in inflammatory diseases [54]. It is well known that regulation of *NF-κB* is a key component of *TNF* signal transduction [53], and NF-κB may be a crucial regulator of inflammation activation and an activator of oxidative stress in goats [15]. Maddox et al. [55] demonstrated that supplementation with Se decreases agonist-induced neutrophil adherence to endothelial cells by increasing GPX concentration and decreasing hydrogen peroxide, IL−1, and TNF-α levels in bovine mammary endothelial cells. Hence, we observed that *TNF*, *TNFSF10*, *TNFSF8*, and *TNFSF13B* gene expression in LD muscle decreases after administration of 4.8 mg/kg SY in growing goats. In short, the above experimental results show that a high level of SY is one of the most important components of the feedstuff that could affect the antioxidant potential and inflammatory response of growing goats. Consistent with our observations, Chauhan et al. [56,57] demonstrated that the feeding of Se could alleviate oxidative stress by regulating the abundances of proinflammatory cytokines and *NF-κB* transcription and *TNF* mRNA abundance in skeletal muscle in Merino × Poll Dorset crossbred ewes.

## 4. Conclusions

In conclusion, the results of the current study show that the inclusion of high SY resulted in 532 DEGs in the tissue of goats, which may be linked with antioxidant and immune functions. High SY has the potential to alleviate oxidative stress in growing goats, because dietary supplementation with 4.8 mg/kg high SY can: (1) improve antioxidant and immune functions, (2) reduce the concentrations of inflammatory molecules in muscle, and (3) modulate antioxidant and inflammatory signalling pathways in growing Qianbei-pockmarked goats.

## 5. Materials and Methods

### 5.1. Experimental Design, Animals, and Diets

The details of the animals and experimental design used in the current study were previously reported by Tian et al. [19]. Briefly, eighteen 4-month-old Qianbei-pockmarked wether goats with similar body weights (25.75 ± 1.75 kg) were randomly divided into 3 groups, with 6 replicates per group, according to the completely randomised design. The control (CON) group was fed a basal diet, and the goats of treatment 1 (LS) and treatment 2 (HS) were fed a basal diet supplemented with 2.4 and 4.8 mg/kg SY, respectively. The animal feeding experiment lasted 74 d (including a preparation period of 14 d and a formal period of 60 d), and feed and water were available *ad libitum* during the entire feeding trial period. SY is a commercial product, and it was obtained from Jiangsu Qianbo Bioengineering Co., Ltd. (Nanjing, Jiangsu, China). The National Research Council report was referred to for the nutrient requirements of goats in this study [58]; the ingredients and nutrient composition of the basal diets are shown in Appendix A.

### 5.2. Chemical Composition

The basal diet was dried at 65 °C in a vacuum oven for 72 h and run through a 1 mm sieve after being ground. The chemical compositions of dry matter (DM; method 930.15), crude protein (CP; N × 6.25; method 988.05), ether extract (EE; method 920.39), neutral detergent fibre (NDF; method 2002.04), acid detergent fibre (ADF; method 973.18), and ash (method 942.05) were determined according to the methods of the Association of Official Analytical Chemists [59].

### 5.3. Apparent Digestibility

The digestion trial was carried out from 68 to 74 d, which includes a preparation period of 2 d and a formal period of 5 d. The apparent nutrient digestibilities were analysed as per the method of the acid-insoluble ash indicator method [60].

### 5.4. Plasma Biochemical and Antioxidant Activity Parameters

The blood sample (10 mL) was collected from the jugular vein of each kid into a vacuette tube with heparin sodium at 73 d. The above plasma was collected via centrifuging at 3000× *g* for 15 min at 4 °C and was stored at −80 °C for further analysis. Glucose (Glu), urea nitrogen (UN), alanine aminotransferase (ALT), aspartate aminotransferase (AST), total cholesterol (TC), triglyceride assay (TG), creatinine (Cr), total protein (TP), albumin (Alb), total antioxidant capacity (TAC), superoxide dismutase (SOD), GPX, reduced glutathione (GSH), glutathione S-transferase (GST), catalase (CAT), and malondialdehyde (MDA) were detected using a microplate reader (Epoch, BioTek, Luzern, Switzerland). Furthermore, the respective commercial assay kits were obtained from a commercial company (Nanjing Jiancheng Bioengineering Institute, Nanjing, China).

### 5.5. Muscle Immune and Oxidation Parameters

The apparent digestibility, plasma biochemicals, and antioxidant activity parameters as well as our previous research results [19] indicate that dietary supplementation of 4.8 mg/kg SY can enhance antioxidant activity and improve muscle fatty acid and amino acid profiles in goats. Hence, all goats in CON and HS were slaughtered for further analysis. The longissimus dorsi (LD) muscle was immediately separated, and LD homogenate was prepared by mixing with phosphate buffered saline (1:9) after centrifuging at 4000× *g* and 4 °C for 10 min; the supernatant was separated and stored at −80 °C. Immunoglobulin A (IgA), immunoglobulin G (IgG), immunoglobulin M (IgM), selenoprotein P (SEPP), interleukin−1 (IL−1), nuclear factor kappa-B (NF-κB), and 8-hydroxy-2-deoxyguanosine (8-OHdG) were analysed using ELISA kits, and all kits were obtained from Jiangsu Meimian Industry Co., Ltd. (Nanjing, Jiangsu, China). Nitric oxide (NO), superoxide anion (O^2−^), and hydroxyl free radicals (OH) were detected using kits purchased from the Nanjing Jiancheng Bioengineering Institute.

### 5.6. Transcriptome

The rumen was immediately separated after slaughter, washed with normal saline, kept in a 1.5 mL tube containing RNA wait solution, and then sent to Shanghai Meiji Biomedical Technology Co., Ltd. (Shanghai, China) for sequencing.

#### 5.6.1. RNA Extraction

Total RNA was extracted from the tissue using TRIzol^®^ reagent according to the manufacturer’s instructions (Invitrogen), and genomic DNA was removed using DNase I (TaKaRa). Then, RNA quality was determined using a 2100 Bioanalyzer (Agilent) and quantified using an ND−2000 (NanoDrop Technologies). Only high-quality RNA samples (OD260/280 = 1.8~2.2, OD260/230 ≥ 2.0, RIN ≥ 6.5, 28S:18S ≥ 1.0, >1 μg) were used in constructing the sequencing library.

#### 5.6.2. Library Preparation and Illumina HiSeq xten/Nova Seq 6000 Sequencing

The RNA-Seq transcriptome library was prepared following the TruSeqTM RNA sample preparation kit from Illumina (San Diego, CA, USA) using 1 μg of total RNA. Briefly, messenger RNA was first isolated according to the polyA selection method via oligo (dT) beads and then fragmented via fragmentation buffer. Next, double-stranded complementary deoxyribonucleic acid (cDNA) was synthesised using a SuperScript double-stranded cDNA synthesis kit (Invitrogen, CA, USA) with random hexamer primers (Illumina). Then, the synthesised cDNA was subjected to end repair, phosphorylation, and “A” base addition according to Illumina’s library construction protocol. Libraries were size selected for cDNA target fragments of 300 bp on 2% low range ultra agarose followed by polymerase chain reaction (PCR) amplification using Phusion DNA polymerase for 15 PCR cycles. After quantification via TBS380, the paired-end RNA-Seq sequencing library was sequenced with the Illumina HiSeq xten/NovaSeq 6000 sequencer (2 × 150 bp read length).

#### 5.6.3. Read Mapping

The raw paired-end reads were trimmed and quality controlled using SeqPrep and Sickle with default parameters. Then, clean reads were separately aligned to the reference genome in orientation mode using HISAT2 software. The mapped reads of each sample were assembled using StringTie in a reference-based approach.

#### 5.6.4. Differential Expression Analysis and Functional Enrichment

To identify DEGs between two different samples, the expression level of each transcript was calculated according to the transcripts per million reads method. RSEM was used to quantify gene abundances. Essentially, differential expression analysis was performed using DESeq2/DEGseq/EdgeR with a Q value ≤ 0.05, and DEGs with |log2FC| > 1 and Q value ≤ 0.05 by DESeq2 or EdgeR) or Q value ≤ 0.001 by DEGseq were considered to be significantly differentially expressed genes. In addition, functional enrichment analysis comprising gene ontology (GO) and KEGG analyses were performed to identify the significantly enriched GO terms and metabolic pathways for the identified DEGs compared with the whole-transcriptome background at a Bonferroni-corrected *p* value ≤ 0.05. GO functional enrichment and KEGG pathway analysis were carried out using Goatools and KOBAS.

#### 5.6.5. Identification of Alternative Splicing Events

All alternative splicing events that occurred in our sample were identified using the recently released program rMATS. Only the isoforms found to be similar to the reference sequence or novel splice junctions were considered, and the splicing differences were detected as exon inclusion, exon exclusion, alternative 5′ end, alternative 3′ end, and intron retention.

### 5.7. Validation of RNA-Seq Results via Real−Time PCR

Seven DEGs related to antioxidant activities were selected for verification using real-time PCR and transcriptome sequencing analysis. All primers were designed using the GenScript real-time PCR (TaqMan) primer and probe design tool and were synthesised by Wuhan Servicebio Technology Co., Ltd. (Wuhan, China). The seven target genes were selenoprotein W, 1 (SEPW1), glutathione peroxidase 1 (GPX1), glutathione S-transferase A1 (GSTA1), tumour necrosis factor (TNF), tumour necrosis factor superfamily member 10 (TNFSF10), tumour necrosis factor superfamily member 8 (TNFSF8), and tumour necrosis factor superfamily member 13b (TNFSF13B). Glyceraldehyde-3-phosphate dehydrogenase (GAPDH) was used as a housekeeping gene. The cDNA synthesis and real-time PCR amplification were performed as follows: Each muscle cDNA was analysed using real-time PCR with a 15 µL reaction volume of 2× SYBR Green qPCR master mix (7.5 µL), 2.5 µM primer (1.5 µL), muscle cDNA (2.0 µL), and distilled water (4.0 µL). The real-time PCR conditions were as follows: predenaturation at 95 °C for 10 min, 40 cycles of denaturation at 95 °C for 15 s, and extension at 60 °C for 30 s. Each muscle cDNA was tested in triplicate.

### 5.8. Statistical Analysis

Each kid was used as an experimental unit (n = 6). The nutrient apparent digestibilities and blood antioxidant activities were analysed using SAS 9.1.3 (SAS Institute, Cary, NC, USA) via one-way analysis of variance, and the least squares mean was reported using Tukey’s test. Relative mRNA gene expression was calculated using the 2^−ΔΔCt^ method, and the average of the CON group was used as a calibrator. The transcriptome sequencing data were analysed using the Majorbio Cloud Platform (www.majorbio.com). The difference was significant when the *p* value was less than 0.05.

## Figures and Tables

**Figure 1 ijms-24-01124-f001:**
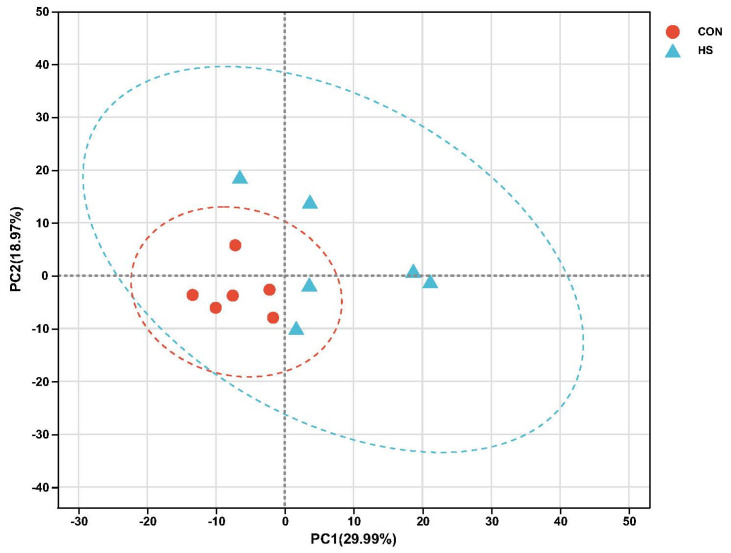
Principal component analysis of the transcriptome profiles.

**Figure 2 ijms-24-01124-f002:**
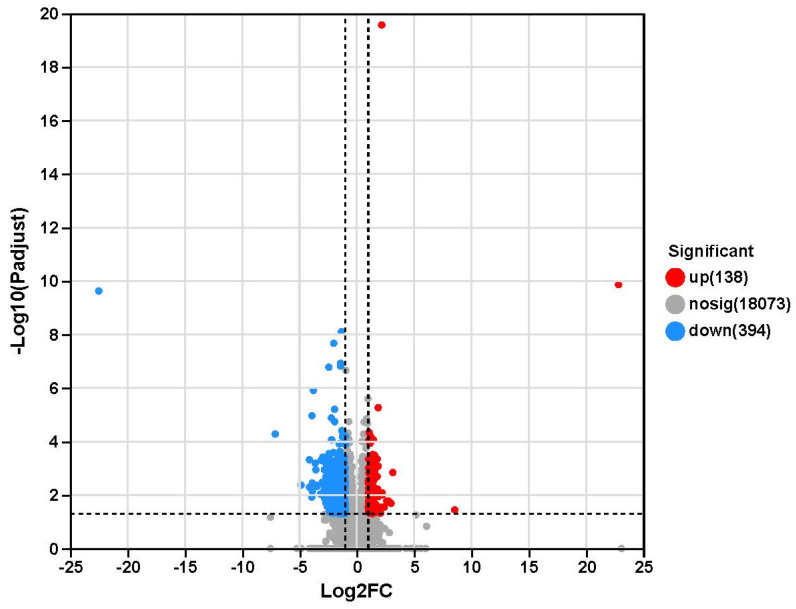
Volcano map plot of differential gene expression. The ordinate is the −log10 *p* adjust value, and the abscissa is the log2FoldChange value. The red colour means up-regulated genes, the blue colour means downregulated genes, and the green colour means genes with no difference.

**Figure 3 ijms-24-01124-f003:**
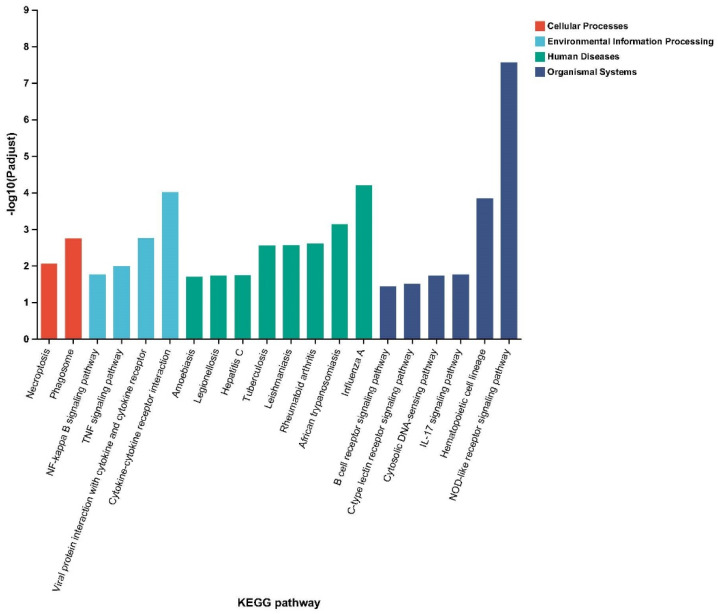
KEGG enrichment analysis histogram. The *x* axis is the KEGG term, and the *y* axis is the log10 *p* adjust value for the KEGG term.

**Figure 4 ijms-24-01124-f004:**
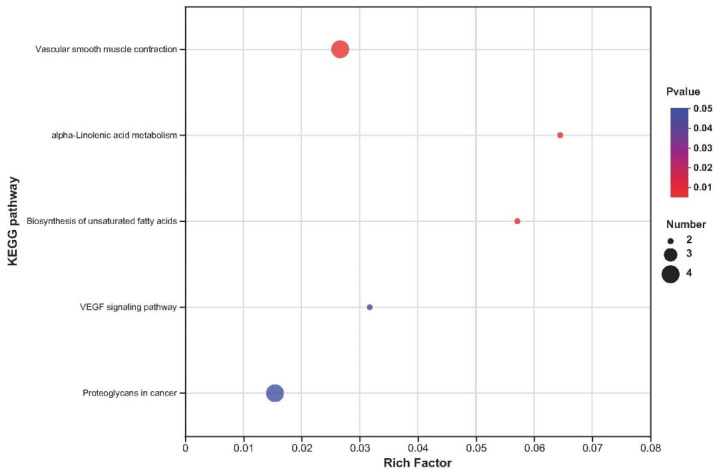
KEGG enrichment analyses of the up-regulated DEGs. The *x* axis represents enrichment factor, and the *y* axis represents pathway category.

**Figure 5 ijms-24-01124-f005:**
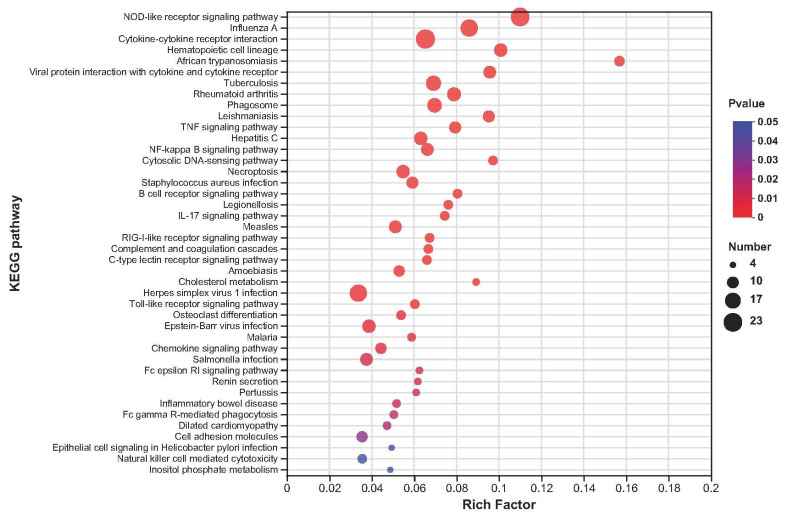
The KEGG enrichment analyses of the down-regulated DEGs. The *x* axis represents enrichment factor, and the *y* axis represents pathway category.

**Figure 6 ijms-24-01124-f006:**
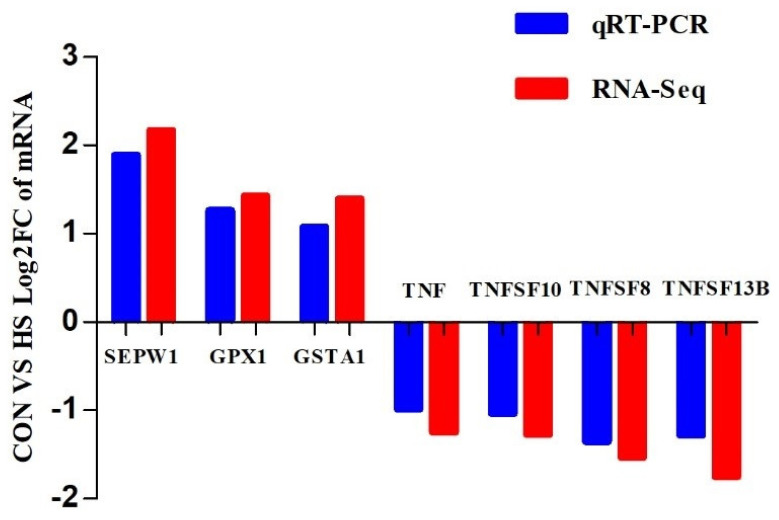
The mRNA expression levels of 7 DEGs between CON and HS. The *x* axis and *y* axis represent the CON vs. HS log2FC of mRNA measured using RNA-Seq and real-time PCR, respectively. *SEPW1*: selenoprotein W, 1; *GPX1*: glutathione peroxidase 1; *GSTA1*: glutathione S-transferase A1; *TNF*: tumour necrosis factor; *TNFSF10*: tumour necrosis factor superfamily member 10; *TNFSF8*: tumour necrosis factor superfamily member 8; *TNFSF13B*: tumour necrosis factor superfamily member 13b.

**Table 1 ijms-24-01124-t001:** Effect of selenium-yeast on nutrient apparent digestibilities of goats.

Item, %		Treatment		SEM	*p*-Value
	CON	LS	HS		
DM	87.69	87.21	87.47	0.3803	0.6865
OM	74.97	77.20	76.32	0.6703	0.1371
GE	74.07	75.38	74.80	0.5289	0.2852
CP	69.77	71.40	68.62	1.0650	0.2555
EE	71.63 ^a^	72.07 ^a^	74.21 ^b^	0.3730	0.0058
NDF	53.28	54.56	57.16	2.7704	0.6250
ADF	48.60	50.67	49.14	1.3658	0.5707

Different letters within a row are significantly different (*p* < 0.05). DM: dry matter; OM: organic matter; GE: gross energy; CP: crude protein; EE: ether extract; NDF: neutral detergent fibre; ADF: acid detergent fibre.

**Table 2 ijms-24-01124-t002:** Effect of selenium-yeast on plasma biochemical and antioxidant activity parameters of goats.

Item		Treatment		SEM	*p*-Value
	CON	LS	HS		
Glu (mmol/L)	4.47	4.74	4.31	0.2052	0.3343
UN (mmol/L)	4.90	5.20	4.94	0.3118	0.7675
ALT (U/L)	5.07	5.24	5.78	0.3941	0.4888
AST (U/L)	31.92 ^a^	25.58 ^b^	20.17 ^c^	1.1854	<0.0001
TC (mmol/L)	4.40	3.91	4.04	0.9733	0.6569
TG (mmol/L)	0.53	0.50	0.55	0.0639	0.8553
Cr (μmol/L)	67.24	64.83	66.45	2.5398	0.8008
TP (gprot/L)	4.20 ^a^	4.35 ^ab^	4.44 ^b^	0.0610	0.0246
Alb (g/L)	24.42	22.58	23.35	0.9950	0.4299
TAC (U/mL)	1.92 ^b^	2.15 ^b^	3.74 ^a^	0.3329	0.0054
SOD (U/mL)	16.94	17.19	17.47	0.6525	0.8475
GPX (U/mL)	330.88	331.37	342.77	6.1516	0.3130
GSH (mg/L)	4.73	5.71	4.31	0.4591	0.7397
GST (U/mL)	42.21 ^a^	64.55 ^b^	57.38 ^ab^	5.1805	0.0403
CAT (U/mL)	1.75 ^a^	2.87 ^b^	2.02 ^ab^	0.3068	0.0355
MDA (nmol/mL)	2.03 ^a^	0.72 ^b^	1.03 ^b^	0.2059	0.0061

Different letters within a row are significantly different (*p* < 0.05). Glu: glucose; UN: urea nitrogen; ALT: alanine aminotransferase; AST: aspartate aminotransferase; TC: total cholesterol; TG: triglyceride; Cr: creatinine; TP: total protein; Alb: albumin; TAC: total antioxidant capacity; SOD: superoxide dismutase; GPX: glutathione peroxidase; GSH: reduced glutathione; GST: glutathione S-transferase; CAT: catalase; MDA: malondialdehyde.

**Table 3 ijms-24-01124-t003:** Effect of selenium-yeast on muscle oxidation products of goats.

Item	CON	HS	SEM	*p*-Value
IgA (μg/mL)	85.86	79.94	4.3429	0.3896
IgG (mg/mL)	1.30	1.28	0.1738	0.9377
IgM (μg/mL)	440.20 ^a^	505.52 ^b^	10.0882	0.0486
SEPP (ng/mL)	8.43	7.56	1.2584	0.6500
IL−1 (pg/mL)	12.61	11.17	1.0881	0.4008
NF-κB (pg/mL)	4.69 ^a^	3.06 ^b^	0.3585	0.0322
8-OHdG (ng/mL)	0.91	0.79	0.0753	0.3301
NO (μmol/L)	20.92	35.23	4.8268	0.1580
O^2−^ (U/L)	201.98 ^a^	181.51 ^b^	4.5181	0.0094
OH (U/mL)	81.29	82.62	0.6687	0.1884

Different letters within a row are significantly different (*p* < 0.05). IgA: immunoglobulin A; IgG: immunoglobulin G; IgM: immunoglobulin M; SEPP: selenoprotein P; IL−1: interleukin−1; NF-κB: nuclear factor kappa-B; 8-OHdG: 8-hydroxy-2-deoxyguanosine; NO: nitric oxide; O^2−^: superoxide anion; ·OH: hydroxyl free radical.

## Data Availability

Sequencing files with raw data associated with each sample have been submitted to the National Center for Biotechnology Information under study accession number PRJNA874455.

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
