# Peer review of "Differentially Expressed Genes and Signalling Pathways Regulated by High Selenium Involved in Antioxidant and Immune Functions of Goats Based on Transcriptome Sequencing"

_ijms, 2023, doi:10.3390/ijms24021124_

Round 1
Reviewer 1 Report
This paper is interesting, it takes the concepts given in the introduction section but the introduction can be improved to give more solid information about the main idea of the paper. However, the management of the information and the experimental design is good. The data can be improved to have a better and more understanding. The conclusion has to be improved to get more interest from the readers to continue researching this topic.
Author Response
Dear reviewer:
Authors would like to express our most sincere gratitude to you for your effort and patience in reviewing our manuscript. We have substantially revised it after reading the comments. Moreover, the red color means answer to your question/suggestions.
This paper is interesting, it takes the concepts given in the introduction section but the introduction can be improved to give more solid information about the main idea of the paper. However, the management of the information and the experimental design is good. The data can be improved to have a better and more understanding. The conclusion has to be improved to get more interest from the readers to continue researching this topic.
Answer: The authors thank for the positive comment. The authors hope to learn more knowledge to you. Firstly, the authors have rewritten introduction section and added some new reference citations in Introduction section according to the suggestion of the reviewer. Secondly, the authors have also improve the conclusion section, please check the line 445-452 in the new manuscript. Thirdly, we employed an English-language editing service, AJE, to polish our wording before submitting this manuscript, and certification is attached at the end of the new manuscript.
Finally, thank you again for your some pertinent comments and we hope to learn more knowledge from where you are.
Reviewer 2 Report
The presented research has scientific novelty. It is performed on an important scientific topic. The most modern methods of genetic analysis were used. Adequate bioinformatics methods have been used. Demonstrated how the inclusion of certain substances provides better antioxidant protection in animals. Which genes are highly expressed in which tissues, how provide this process.
For a better understanding of the results, a few questions.
1. Explain why exactly 2.4 and 4.8 mg/kg SY was added to the basal diet, for example, not more or less.
2. In the results of study 3.2. Plasma Biochemical and Antioxidant Activity Parameters it is desirable to include a comparison on GST between control (CON), treatment 1 (LS) and treatment 2 (HS) groups.
3. It is desirable to indicate which bioethical principles were used by the authors in slaughtering animals.
4. Explain why animals from the control group (CON) and treatment 2 (HS) were slaughtered and gene expression in the muscle tissue of these groups was studied accordingly. Why treatment group 1 (LS) was excluded.
5. The experiment was conducted over 74 days. Dietary supplementation with 4.8 mg/kg SY was shown to potentially alleviate oxidative stress by improving antioxidant and immune function, reducing inflammation and modulating antioxidant and inflammatory signaling pathways in growing goats. Were there differences between the animals in body weight or other productivity traits. Because it can be assumed that better health provides more growth energy for the goats.
Author Response
Dear reviewer:
Authors would like to express our most sincere gratitude to you for your effort and patience in reviewing our manuscript. We have substantially revised it after reading the comments. Moreover, the red color means answer to your question/suggestions.
The presented research has scientific novelty. It is performed on an important scientific topic. The most modern methods of genetic analysis were used. Adequate bioinformatics methods have been used. Demonstrated how the inclusion of certain substances provides better antioxidant protection in animals. Which genes are highly expressed in which tissues, how provide this process.
Answer: The authors thank for the positive comment. The authors hope to learn more knowledge to you.
For a better understanding of the results, a few questions.
1. Explain why exactly 2.4 and 4.8 mg/kg SY was added to the basal diet, for example, not more or less.
Answer: Various previous studies have proven that ruminants have a high tolerance for Se because Se is less available to ruminants than to nonruminants, and the maximum requirement for Se in ruminant diets is 5-8 mg/kg. However, the molecular mechanism through which high levels of SY impact the muscle antioxidant activity of growing goats is still unknown. Therefore, the objective of this study was to observe the effect of high SY on antioxidant and immune functions of growing goats based on transcriptome sequencing. In addition, the SY as part of the basal ration and was mixed in the concentrate, and then the concentrate was mixed with the roughage evenly to make sure accuracy.
2. In the results of study 3.2. Plasma Biochemical and Antioxidant Activity Parameters it is desirable to include a comparison on GST between control (CON), treatment 1 (LS) and treatment 2 (HS) groups.
Answer: The authors agreed with the reviewer, the authors have added the content in the line 227-229 in the new manuscript.
3. It is desirable to indicate which bioethical principles were used by the authors in slaughtering animals.
Answer: The slaughtering animals were cared for and handled in accordance with the Experimental Animal Ethics Committee of Guizhou University (approval number: EAE-GZU-2020-7009). The authors have added it and please check the line 469-471 in the new manuscript.
4. Explain why animals from the control group (CON) and treatment 2 (HS) were slaughtered and gene expression in the muscle tissue of these groups was studied accordingly. Why treatment group 1 (LS) was excluded.
Answer: Thank you for your professional suggestion. According to the apparent digestibility (Table 1), plasma biochemical and antioxidant activity parameters (Table 2), and our previous research results [19] indicated that dietary supplementation of 4.8 mg/kg SY can enhance antioxidant activity, and improve muscle fatty acid and amino acid profiles in goats. Hence, all goats in CON and HS were slaughtered for further analysis. Thus, we select CON and HS observe the effect of high selenium on antioxidant function of growing goats based on transcriptome sequencing in this study.
5. The experiment was conducted over 74 days. Dietary supplementation with 4.8 mg/kg SY was shown to potentially alleviate oxidative stress by improving antioxidant and immune function, reducing inflammation and modulating antioxidant and inflammatory signaling pathways in growing goats. Were there differences between the animals in body weight or other productivity traits. Because it can be assumed that better health provides more growth energy for the goats.
Answer: Thank you for your professional suggestion. In our previous study, we found that adding 4.8 mg/kg SY had no significant influence on the dry matter intake, and average daily gain, whereas it can improve meat quality, muscle fatty acid and amino acid profiles in Qianbei-pockmarked goats (Tian et al., 2022; doi: 10.3389/fvets.2021.813672). The authors just detected body weight, and did not observe other productivity traits. In this manuscript, we observe the effect of high SY (4.8 mg/kg) on antioxidant and immune functions of growing goats.
Finally, thank you again for your some pertinent comments and we hope to learn more knowledge from where you are.
Reviewer 3 Report
General comments:
This study mainly aimed to evaluate gene expression to improve antioxidant activity in goats challenged by two levels of selenium-yeast. The experimental design is related to a previous study recently published [17] by the authors. The manuscript is very well written and full discussed. Some minor suggestions/comments were made (see specific section) mainly related to the presentation of the results.
Specific comments:
L12: “… selenium-yeast (SY), respectively.”
L56: “…reticulum and rumen…”
L90: I suggest to add the mean age of these kids (it was not reported in your previous published study [17]).
L206: What is the (post-hoc) test used to evaluate differences between pairs?
L210: “…between the three groups…”.
L211: TP content is not reported in Table 2. Start the first (left) column with “a” superscript letter. Check all tables.
L218: Please define all abbreviations in the legend. Check all tables.
L228: Why the LS treatment was not inserted. The heading is lacking in Table 3.
L72: I suggest to identify the significance in Figure 5 such as reported in text.
L381: “This is because…”?
L383: Please, rewrite the citation.
Author Response
Dear reviewer:
Authors would like to express our most sincere gratitude to you for your effort and patience in reviewing our manuscript. We have substantially revised it after reading the comments. Moreover, the red color means answer to your question/suggestions.
General comments:
This study mainly aimed to evaluate gene expression to improve antioxidant activity in goats challenged by two levels of selenium-yeast. The experimental design is related to a previous study recently published [17] by the authors. The manuscript is very well written and full discussed. Some minor suggestions/comments were made (see specific section) mainly related to the presentation of the results.
Answer: The authors thank for the positive comment. The authors hope to learn more knowledge to you.
Specific comments:
L12: “… selenium-yeast (SY), respectively.”
Answer: Thank you for polishing English for us. We have revised it according to your suggestion. Please check the new manuscript.
L56: “…reticulum and rumen…”
Answer: Thank you for polishing English for us. We have revised it according to your suggestion. Please check the line 59 in the new manuscript.
L90: I suggest to add the mean age of these kids (it was not reported in your previous published study [17]).
Answer: Thank you for your professional suggestion. The experimental kids is 4 months old, the authors have added in line 96 of the new manuscript.
L206: What is the (post-hoc) test used to evaluate differences between pairs?
Answer: Thank you for your professional suggestion. The nutrient apparent digestibilities were analysed using SAS 9.1.3 (SAS Institute, Cary, NC, USA) via one-way analysis of variance, and least squares mean was reported using the Tukey’s test. The authors have added related content, and please check the line 205-212 in the new manuscript.
L210: “…between the three groups…”.
Answer: Thank you for polishing English for us. We have revised it according to your suggestion, and we also checked the entire manuscript and made corresponding modifications. Please check the new manuscript.
L211: TP content is not reported in Table 2. Start the first (left) column with “a” superscript letter. Check all tables.
Answer: We are very sorry we forget TP and Alb parameters in Table 2, and the author have added them in Table 2. Moreover, the authors have revised the superscript letter according to your suggestion. Please check all tables in the new manuscript.
L218: Please define all abbreviations in the legend. Check all tables.
Answer: Thank you for your professional suggestion. The authors agreed with the reviewer, and we had defined all abbreviations in the legend for all tables in the new manuscript.
L228: Why the LS treatment was not inserted. The heading is lacking in Table 3.
Answer: Thank you for your professional suggestion. According to the apparent digestibility (Table 1), plasma biochemical and antioxidant activity parameters (Table 2), and our previous research results indicated that dietary supplementation of 4.8 mg/kg SY can enhance antioxidant activity, and improve muscle fatty acid and amino acid profiles in goats. Hence, all goats in CON and HS were slaughtered for further analysis. Please check the line 129-134 in the new manuscript.
L72: I suggest to identify the significance in Figure 5 such as reported in text.
Answer: Thank you for your professional suggestion. We obtained 532 DEGs for transcriptome sequencing between CON and HS (Figure 2; Table S3). Moreover, a total of 138 upregulated genes (p < 0.05) and 394 downregulated genes (p < 0.05) were identified in the HS group relative to the CON group. In fact, we used significant 532 DEGs for further analysis, and we found that the downregulated DEGs were enriched in 231 KEGG pathways, and 42 KEGG pathways mainly related to the NOD-like receptor signalling pathway, influenza A, cytokine‒cytokine receptor interaction, haematopoietic cell lineage, and African trypanosomiasis were significantly enriched (p < 0.05). In fact, these pathways and candidate genes are significant data for all figures. Please check the line 259-262, line 274-281 in the new manuscript.
L381: “This is because…”?
Answer: The authors have rewritten the sentence according to your suggestion, please check the line 398-400 in the new manuscript.
L383: Please, rewrite the citation.
Answer: The authors have added the citation in the new manuscript, please check the line 400 in the new manuscript.
Finally, thank you again for your some pertinent comments and we hope to learn more knowledge from where you are.